# Total Pelvic Exenteration, Cytoreductive Surgery, and Hyperthermic Intraperitoneal Chemotherapy for Rectal Cancer with Associate Peritoneal Metastases: Surgical Strategies to Optimize Safety

**DOI:** 10.3390/cancers12113478

**Published:** 2020-11-23

**Authors:** Jean-Jacques Tuech, Jean Pinson, François-Xavier Nouhaud, Gregory Wood, Thomas Clavier, Jean-Christophe Sabourin, Frederic Di Fiore, Matthieu Monge, Eloïse Papet, Julien Coget

**Affiliations:** 1Department of Digestive Surgery, Rouen University Hospital, 1 rue de Germont, F-76031 Rouen, France; jean.pinson@chu-rouen.fr (J.P.); Matthieu.Monge@chu-rouen.fr (M.M.); eloise.papet@chu-rouen.fr (E.P.); julien.coget@chu-rouen.fr (J.C.); 2Department of Urology, Rouen University Hospital, 1 rue de Germont, F-76031 Rouen, France; Francois-Xavier.Nouhaud@chu-rouen.fr; 3Department of Anesthesiology and Critical Care, Rouen University Hospital, 1 rue de Germont, F-76031 Rouen, France; Gregory.Wood@chu-rouen.fr (G.W.); thomas.clavier@chu-rouen.fr (T.C.); 4Department of Pathology, Iron Group, Rouen University Hospital, 1 rue de Germont, F-76031 Rouen, France; Jean-Christophe.Sabourin@chu-rouen.fr; 5Department of Digestive Oncology, Rouen University Hospital, 1 rue de Germont, F-76031 Rouen, France; Frederic.DiFiore@chu-rouen.fr

**Keywords:** cytoreductive surgery, hyperthermic intraperitoneal chemotherapy, pelvic exenteration, ileal conduit, urinary leakage, empty pelvis syndrome, peritoneal metastases, peritoneal carcinomatosis

## Abstract

**Simple Summary:**

Cytoreductive surgery (CRS) with hyperthermic intraperitoneal chemotherapy (HIPEC) is a curative treatment for patients with peritoneal carcinomatosis. Pelvic exenteration is an established treatment option for locally advanced pelvic malignancy. Based on the argument that high-risk complications arise from each procedure, the majority of researchers do not recommend performing a CRS/HIPEC with pelvis exenteration. Herein, we critically analyzed the data from 16 patients treated by these two procedures for 15 rectal and one appendiceal adenocarcinomas. Clear resection (R0) margins were achieved in 81.2% of cases. The median hospital stay was 46 days (26–129), and nine patients (56.2%) experienced severe complications that led to death in two cases (12.5%). Survival rates were not clarified, since the follow-up is ongoing. Pelvis exenteration associated with CRS/HIPEC may be a reasonable procedure in selected patients at expert centers. Pelvic involvement should not be considered a definitive contraindication for CRS/HIPEC if a R0 resection could be achieved. However, the morbidity and the mortality are high with this combination of treatment, and further research is needed to assess the oncologic benefit and quality of life before such a radical approach can be recommended.

**Abstract:**

Background: Cytoreductive surgery (CRS) with hyperthermic intraperitoneal chemotherapy (HIPEC) is a curative treatment option for patients with peritoneal carcinomatosis. Total pelvic exenteration (TPE) is an established treatment option for locally advanced pelvic malignancy. These two procedures have high mortality and morbidity, and therefore, their combination is not currently recommended. Herein, we reported our experience on TPE associated with CRS/HIPEC with a critical analysis for rectal cancer with associate peritoneal metastases. Methods: From March 2006 to August 2020, 319 patients underwent a CRS/HIPEC in our hospital. Among them, 16 (12 men and four women) underwent an associated TPE. The primary endpoints were perioperative morbidity and mortality. Results: There was locally recurrent rectal cancer in nine cases, six locally advanced primary rectal cancer, and a recurrent appendiceal adenocarcinoma. The median Peritoneal Cancer Index (PCI) was 8. (4–16). Mean duration of the surgical procedure was 596 min (420–840). Complete cytoreduction (CC0) was achieved in all patients, while clear resection (R0) margins on the resected pelvic organs were achieved in 81.2% of cases. The median hospital stay was 46 days (26–129), and nine patients (56.2%) experienced severe complications (grade III to V) that led to death in two cases (12.5%). The total reoperation rate for patients was 6/16 (37.5%) and 3/16 (18.75%) with percutaneous radiological-guided drainage. Conclusions: In summary, TPE/extended TPE (ETPE) associated with CRS/HIPEC may be a reasonable procedure in selected patients at expert centers. Pelvic involvement should not be considered a definitive contraindication for CRS/HIPEC in patients with resectable peritoneal surface diseases if a R0 resection could be achieved on all sites. However, the morbidity and the mortality are high with this combination of treatment, and further research is needed to assess the oncologic benefit and quality of life before such a radical approach can be recommended.

## 1. Introduction

Cytoreductive surgery (CRS) with hyperthermic intraperitoneal chemotherapy (HIPEC) is a curative treatment option for patients with peritoneal carcinomatosis (PC). It has a long-term survival benefit [1,2] but is associated with high rates of morbidity, ranging from 12% to 65% [3,4,5,6]. Pelvic exenteration is an established treatment option for locally advanced primary rectal cancer (LARC) and locally recurrent rectal cancer (LRRC), but it is associated with significant morbidity; therefore, many investigators do not recommend CRS-HIPEC with pelvic exenteration. Since the majority of researchers consider it as an exclusion criterion, only a few case reports [7,8,9] have been published to date. Herein, we reported our experience with total pelvic exenteration (TPE) associated with CRS and HIPEC with a critical analysis. We described the strategies developed over time to reduce the mortality and morbidity of this association in order to make it a safe surgical approach.

## 2. Results

### 2.1. Patients Characteristics

From March 2006 to August 2020, 319 patients underwent a CRS and HIPEC in our hospital. Among them, 16 (12 men and four women) underwent an associated TPE or extended TPE (ETPE). The baseline patient characteristics are reported in Table 1. The main indication was rectal cancer in 15 cases: locally recurrent rectal cancer (LRCC) in nine cases and six locally advanced primary rectal cancer (LARC) cases. The remaining patient was operated on for a recurrent appendiceal adenocarcinoma. Next, 12 patients who had previously undergone pelvic surgery: 11 proctectomies and one posterior exenteration were already treated with HIPEC 77 months before ETPE.

### 2.2. Peroperative Data

Detailed information on the surgical resections is reported in Table 2. The median PCI was 8 (4–16); this calculation did not take into account the pelvic region; the median number of regions affected by peritoneal carcinoma was three (two–nine), and the pelvic region was also excluded for this calculation. The mean duration of the surgical procedure was 596 min (420–840). Six patients had TPE and 10 had ETPE. The lateral compartment required resection on both sides in five cases and on one side in four cases. The posterior compartment required resection in nine cases. Urinary reconstruction used an ileal conduit (Bricker procedure) in 10 cases, and bilateral ureterostomy was constructed in six cases. The empty pelvis was managed in 11 cases using different methods, but the filling was systematically carried out with patient 7. Complete cytoreduction (CC0) was achieved in all patients, while clear resection (R0) margins on the resected pelvic organ was achieved in 81.2% of cases. Three LRCC patients had R1 resection. The involved margin was on the posterior compartment despite the resection of the pre-sacral fascia.

### 2.3. Postoperative Morbidity

The details of the postoperative course, the complications, the time to the first recurrence, and the follow-up are given in Table 3. The median hospital stay was 46 days (26–129), and nine patients (56.2%) experienced severe complications (grade III to V), which led to death in two cases (12.5%). The total reoperation rate for the patients was 6/16 (37.5%) and 3/16 (18.75%) with percutaneous radiological-guided drainage. The most common causes of surgical complications were urosepsis and pelvic abscess. Seven patients died during follow-up: in six cases, the cause of death was secondary to the progression of the oncological disease; patient 1 died from septic shock during chemotherapy treatment (the infection originated from the totally implanted venous access).

## 3. Discussion

Pelvic exenteration for LARC or LRRC remains a surgical challenge associated with high mortality and significant morbidity [10,11,12,13]. Over the past two decades, there has been significant improvement in outcomes and survival rates. The expert center’s reports have shown a five-year survival rate of 36% to 46%, with a mortality rate of 0.6% and a perioperative morbidity rate of 27% [14]. The combination of complete CRS and HIPEC is gradually becoming the standard of care for patients with PC. This approach has been associated with an increased risk of surgical complications, due to the complexity of extensive surgery with multiple intra-abdominal organ resections and peritonectomies. Based on the argument of a high risk of complications from each procedure, the majority do not recommend performing a CRS and HIPEC with TPE or ETPE. To our knowledge, this series is the largest to report specifically on this association to treat patients with LARC or LLRC with associated peritoneal metastases.

Achieving clear margins with an R0 resection has been shown to be the best predictor of long-term survival. Surgical exploration must be complete and meticulous. The presence of an unresectable disease in the abdomen and peritoneum would be a contraindication to perform a TPE. To perform such an extensive resection and have a residual disease at another site would defy the established oncological surgical principles.

In the present series, CCR0 could be achieved in all cases and R0 in 81% (13/15). This rate of R0 resection is consistent with the literature. Denost et al. [15], in an international benchmark trial of the management of LARC and LRRC in France and Australia between 2015 and 2017, demonstrated that the R0 resection rate was lower in France (61.3%) than Australia (91.6%). In our three R1 cases, the involved margin was the posterior one, despite a resection of the pre-sacral fascia. In the future, a more posterior plan (subcortical sacrectomy) could be discussed in this situation in order to obtain an R0 margin. This subcortical sacrectomy could be performed after the removal of the main specimen, with the aim of making this sacrectomy easier. The risk of tumor cell dissemination due to a non-en bloc resection could be managed by HIPEC.

Postoperative complications after major surgical procedures have a negative impact on long-term survival [16,17,18,19,20]; therefore, the reduction of postoperative complications is essential for optimal short- and long-term outcomes. Major complications after CRS/HIPEC in established centers have ranged from 12% to 52% [21]. In the present study, major complications (Dindo > 3) occurred in 50%. Among the independent predictors of major complications, several studies have consistently identified the extent of the disease reflected by the number of organs resected and the duration of surgery [22,23,24]. Surgeons have progressed along a constant learning curve, and the management of peritoneal malignancies [25,26] and expected long-term survival associated with extensive CRS have improved, so expert teams have progressively expanded the indications for surgery, with acceptable morbidity rates [27,28,29]. CRS has been limited to a subset of selected patients likely to tolerate aggressive management [27]. Given the importance of complete CRS in optimizing oncological outcomes, Wagner et al. reported an approach of “cytoreduction at all costs” in appendiceal carcinomatosis [30]. Overall and major complications occurred, respectively, in 70% and 32% of patients after extensive CRS. In our study, patients underwent a median of five resections (range, four–seven), and, according to the definition of Wagner et al. [30], all patients underwent extensive CRS (>three organ resections or >two enteric anastomoses).

Several authors have shown that the rate of postoperative complications increased in cases of CRC-HIPEC associated with urinary system intervention [31,32,33], despite the fact that these studies report only limited urinary resections. To the best of our knowledge, there is no study in the literature investigating the management of urinary tract reconstruction following TPE in the field of CRS and HIPEC procedures. Urologic leaks from a newly formed conduit are a considerable source of morbidity following TPE [34,35,36], leading to prolonged in- and outpatient management, as well as a shorter median survival [35]. Teixeira et al. [35] reported a 16% urine leak rate following TPE. In our study, four patients (25%) experienced a urine leak, which led to death in one case. Ongoing sepsis due to a urine leak is an unfavorable prognostic indicator similar to anastomotic bowel leaks. Brown et al. [34] compared 98 patients who underwent a cystectomy to 133 who underwent a cystectomy as part of a TPE procedure. Postoperative urological complications occurred in 33% of the cystectomy alone group and 59% of the PE group (*p* < 0.001). Urological leaks occurred in 3%, 6%, and 14% of patients who had cystectomy alone, TPE for primary malignancy, and TPE for recurrence, respectively. In the multivariate analysis, more than 5000-mL intraoperative blood loss and previous pelvic radiotherapy independently predicted conduit-associated complications in TPE patients (*p* = 0.002 and 0.035). In our study, nine patients (56%) belonged to this high-risk group (previous radiotherapy, major hemorrhage) of urinary leaks. It should not be overlooked that an ileal conduit led to the confection of three anastomoses. These two points, in order to reduce complications, led us to perform bilateral ureterostomies in five patients (31.2%) rather than an ileal conduit. A recent publication showed a significant reduction of complications [37] when a ureterostomy rather than an ileal conduit was fashioned (Nicola Longo) with an impaired quality of life (QOL). Arman et al. [38] compared patients’ QOL with single stoma cutaneous ureterostomy (SCU), bilateral standard cutaneous ureterostomy (BCU), and an ileal conduit (IC). The IC was associated with better quality of life scores compared to BCU and similar scores compared to SCU. In the future, SCU could be an alternative to BCU in HIPEC patients, but this procedure mobilizes the left ureter more widely to allow its transfer to the right side, which can lead to ischemic damage to the distal ureter. 

One of the major causes of postoperative complications following TPE/ETPE is empty pelvis syndrome. Empty pelvis syndrome can be defined as an empty space or cavity following pelvic exenteration, which may result in fluid accumulation within the pelvic cavity, potentially increasing the risk of pelvic abscess, perineal fluid discharge with perineal wound dehiscence, and prolonged ileus. The irradiated small bowel loops (with an enterostomy following ileal conduit formation) may become adherent to the exposed pelvic surfaces, leading to bowel obstruction and the development of entero-perineal fistulas. This occurs in up to 15% of patients following exenteration, conferring a mortality rate close to 50% [39]. Several methods have been proposed to fill the pelvis and keep the small bowel out of the pelvis, such as breast prosthesis, yet there is a concern regarding prosthesis infection, Cecal pelvic transposition, a myocutaneous flap, and, more recently, implantation of degradable synthetic mesh [40,41,42,43]. In our study, the empty pelvis was managed in 11 cases—systematically, from the seventh patient—using different methods. Delayed coloanal anastomosis (six cases) was privileged when a supra-levator resection was performed. A rectus abdominis muscular flap was used in four cases, a cecal transposition in three cases, an omentoplasty in one, a biological prosthesis in one, and a breast prosthesis in one. In four cases, a combination of two methods was used to completely fill the pelvis. In patient four, we used an omentoplasty and a cecal transposition to fill the pelvis. An omentectomy was mandatory during cytoreductive surgery. We chose this option, because the omentum was macroscopically normal and the PCI was 4. In a recent publication, Bonnefoy et al. [44] demonstrated that, among the 96 patients who underwent a complete cytoreductive surgery with no macroscopic evidence of disease in the greater omentum during surgical exploration, 17 patients (17.70%) had microscopic evidence of a tumor in the omentum. We assumed that, if invisible cancer deposits were present at the surface of the omentum, they would be treated by HIPEC. During follow-up, this patient did not experience recurrence at the omentoplasty level.

Finally, optimizing the patient before multiorgan resection is vital for reducing perioperative morbidity and requires a multi-specialist approach [45,46,47]. In our department, we improved the nutritional status and organized a physical rehabilitation by a physiotherapist for each patient before surgery. However, this was not enough. In the future, all areas for potential improvement must be identified and improve. Formal cardiopulmonary testing is an objective test to assess the fitness and diagnose cardiovascular and lung pathophysiology [48]. A management plan can then be determined for the patient’s perioperative care and pathway [49].

This study has several limitations. The sample size was small, and patients were enrolled for a long period of time, and there has been considerable progress in the perioperative management. Moreover, the quality of life was not assessed by means of dedicated questionnaires or assessments. Furthermore, our survival rates have not yet been clarified, since the follow-up is ongoing. However, despite these weaknesses, our study includes a homogenous group of patients in a single center, which provides useful insight for the challenging surgical strategy.

## 4. Materials and Methods

### 4.1. Patients and Inclusion Criteria

A review of a prospectively maintained database was undertaken to identify and assess the outcomes of patients who underwent TPE, CRS, and HIPEC in the Department of Digestive Surgery, Rouen University Hospital, Rouen, France.

All patients undergoing TPE and HIPEC for any primary or recurrent pathology were included. The diagnosis and management of all malignancies was based on preoperative radiology (CT scan for chest, abdomen, and pelvis; MRI for pelvis; and FDG-PET. MRI for liver, where indicated) and clinical assessment. Indication for an associated HIPEC to TPE was discussed preoperatively for the available preoperative data. A preoperative nutrition assessment was performed, and nutrition support was provided pre- and postoperatively for all patients. The patency for both deep inferior epigastric vessels were analyzed using computed tomography angiography to anticipate the use of a vertical rectus abdominis myocutaneous flap for pelvic filling or perineal reconstruction. All patients received a preoperative mechanical bowel preparation, and preoperative antibiotherapy was given according to local protocols.

### 4.2. Total Pelvic Exenteration and Other Definitions

The surgical principle of pelvic exenteration is a complete en bloc removal of all viscera or structures contiguously involved by tumors with a clear resection margin (R0 resection). To be classified as an R0 resection, a clear margin of >1 mm is required in the histopathological evaluation. Different classifications have evolved to describe different types of recurrence and exenteration; however, there is no universally accepted terminology. In fact, no classification will reflect with accuracy the possible varieties of exenteration procedures, because every procedure is different for every patient.

We used Magrina’s classification [50,51] in order to define the TPE. The TPE is an en bloc resection for pelvic organs, including the internal reproductive organs, bladder, and rectosigmoid. In the presence of upper lesions, adequate tumor resection can be obtained by dividing the viscera above or at the level of the levator muscles. In this procedure, the levator muscle, anus, and urogenital diaphragm are preserved. During low lesion, an infra-levator TPE requires a tailored resection of the levator muscles, urogenital diaphragm, anus, and perineal tissues. Extended TPE (ETPE) is a procedure requiring an additional resection of tissues (small bowel, bone, vessels, etc.). For ETPE, we used the classification proposed by Georgiou et al. [52]. TPE can be enlarged to the posterior compartment (coccyx, pre-sacral fascia, retro-sacral space, and sacrum up to the upper level of S1) or to the lateral compartment (external and internal iliac vessels, lateral pelvic lymph nodes, sciatic nerve, sciatic notch, S1 and S2 nerve roots, and the piriformis or obturator internus muscle).

### 4.3. Cytoreduction and HIPEC 

CRS included the primary tumor removal, complete resection of the tumor nodule with intestinal resection, and peritonectomy. The extent of the peritoneal spread was assessed using the Peritoneal Cancer Index (PCI) [53,54]. The completeness of the cytoreduction (CCR) score was evaluated for each patient before performing HIPEC [55]. CC0 implied no residual macroscopic disease. CC 1, 2, and 3 implied residual disease less than 2.5 mm and 2.5 mm, as well as 2.5 cm and greater than 2.5 cm, respectively. CC 0/1 was macroscopically considered a complete resection, with the subsequent administration of HIPEC. CC 2/3 cases were deemed incomplete cytoreduction, and patients were not given HIPEC. HIPEC was conducted using the open abdominal “coliseum” technique. The technique for CRS/HIPEC has been described elsewhere and was based on Sugarbaker’s principles [55,56]. When the hyperthermic perfusion reached a steady state of 42 °C, the intraperitoneal drug was added to the perfusion. We mainly used two different chemotherapeutic regimens for intraperitoneal perfusion: mitomycin C or oxaliplatin. HIPEC was delivered with 35-mg/m^2^ mitomycin C over a 60-min period or with 460-mg/m^2^ oxaliplatin over a 30-min period. One hour before starting the HIPEC procedure with oxaliplatin, folinic acid 20 mg/m^2^ and 5-fluorouracil 400 mg/m^2^ (in 250-mL saline) were intravenously administered to enhance the effect of oxaliplatin. 

All anastomosis were fashioned after the completion of HIPEC, including urinary reconstruction. Monitoring the urine output during the whole procedure and particularly during HIPEC was an important point for this purpose: bilateral urinary catheters were inserted after the transection of the ureter.

### 4.4. Study Criteria

Standardized clinical data on consecutive patients who underwent TPE/CRS/HIPEC were retrospectively retrieved from prospectively maintained databases. The following preoperative variables were recorded: the demographic characteristics, primary tumor site and histology, comorbidities, history of abdominal surgery, preoperative nutritional status, American Society of Anesthesiologists (ASA) preoperative score, and Eastern cooperative oncology group (ECOG) performance status. Surgery-specific data were collected, including the extent of peritoneal carcinomatosis, extent of TPE, number and type of resected organs, CCR score, duration of surgery, estimated blood loss (EBL), and red blood cell (RBC) transfusion. Clinical outcomes and postoperative complications were recorded, including the incidence of overall complications. The need for reoperation, postoperative length of stay, and mortality were also recorded. All complications were classified according to the Clavien–Dindo classifications [57], which define severe complications by a score of 3 or more. The criterion for complications and operative mortality occurred within 90 days of surgery or at any time during the postoperative hospital stay.

Patients were followed up with, and all were reviewed at one month and then every four months with a physical examination, carcinoembryonic antigen CEA level measurements, and abdominal ultrasonography or a thoracoabdominal CT scan. Local recurrence was defined as a radiologically and/or a biopsy-proven tumor within the pelvis. Distant recurrence was defined as radiologic evidence of a tumor in any other area.

### 4.5. Ethics

The current study was performed with the approval of the institutional ethics committee review board (E2020-72). The specific written informed consent of patients was not required for this observational consecutive case study.

## 5. Conclusions

In summary, TPE/ETPE associated with CRS/HIPEC may be a reasonable procedure in selected patients at expert centers. Pelvic involvement should not be considered a definitive contraindication for CRS/HIPEC in patients with resectable peritoneal surface disease if a R0 resection could be achieved on all sites. A longer follow-up period should make it possible to assess the oncological benefits more surely. The morbidity and the mortality are high with this combination of treatments, and further research is needed to assess the oncologic benefits and quality of life before such a radical approach can be recommended.

## Figures and Tables

**Table 1 cancers-12-03478-t001:** Baseline patient characteristics.

Patient	Year HIPEC	Gender	Age (Y)	BMI	ASA	ECOG PS	Delay/Primary Tumor	Radiochemotherapy (Primary Lesion)	Previous Surgical History	ACE	Neoadjuvant Treatment
1	03/2010	F	61	27.6	2	0	12 months Rectal adenocarcinoma	no	Hysterectomy March 2008 Proctectomy Nov 2008	59	no
2	11/2011	F	46	28	2	0	14 months Rectal adenocarcinoma	no	Proctectomy July 2010 Ovariectomy July 2011 with packing for bleeding	NA	Folfirinox
3	05/2012	M	51	23	2	1	8.5 months Rectal adenocarcinoma	no	R2 Low Hartman sept 2011 Rectal stump leakage	1	Cap 50
4	09/2013	F	58	22	2	0	77 months Appendiceal carcinoma with carcinomatosis PCI 18	no	Right colectomy, posterior pelvectomy, HIPEC oxaliplatin	7	no
5	02/2014	M	69	29.7	2	0	56 months Rectal adenocarcinoma	no	Proctectomy 6/2009	30	Folfox avastin
6	05/2015	M	69	26	2	0	27 months Rectal adenocarcinoma (leak, fecal peritonitis)	Y (2012) Cap 50	Proctectomy 02/2013	2	Folfox 4 vectibix
7	06/2016	M	41	17	2	0	6 months Colon and rectal adenocarcinoma (Lynch Syndrome)	no	Right colectomy 04/2014 Left colectomy 09/2015 Proctectomy (R2) 01/2016	NA	no
8	01/2017	M	44	18	2	0	17 months rectal adenocarcinoma	no	Proctectomy + partial cystectomy 07/2015	21	Folfiri Erbitux
9	02/2018	M	58	21	2	0	Primary rectal adenocarcinoma	no	Explorative laparotomy 07-2017	6	folfox
10	04/2018	M	57	22	2	0	Primary rectal adenocarcinoma	no	Explorative laparotomy 05-2017 Explorative laparotomy 10-2017	NA	Folfirinox Cap 50
11	01/2019	M	39	29	1	0	24 months Rectal adenocarcinoma	no	Proctectomy 01-2017		folfiri
12	03/2019	M	45	32.5	1	0	Primary rectal adenocarcinoma (signet ring cell)	no	none	3	folfirinox
13	03/2020	H	55	20	1	0	Primary rectal adenocarcinoma	no	none	4.6	Cap 50
14	05/2020	M	67	25	2	0	18 months Rectal adenocarcinoma	no	proctectomy + partial cystectomy 01-2019	4	Folfox
15	06/2020	M	65	26	2	0	Rectal adenocarcinoma 24 months	Y (2018) Cap 50	Proctectomy 06-2018	11	Folfiri avastin
16	07/2020	F	66	27	2	0	Rectal adenocarcinoma 8 months	no	Proctectomy 12-2019	14	Folfiri avastin

HIPEC: hyperthermic intraperitoneal chemotherapy, M: male, F: female, BMI: Body Mass Index, ASA: American Society of Anesthesiologists preoperative score, ECOG PS: Eastern Cooperative Oncology Group-Performance Status, PCI: Peritoneal Cancer Index, and Cap 50: 50 Gy irradiation and 1600-mg/m oral capecitabine.

**Table 2 cancers-12-03478-t002:** Detailed information on the surgical resections.

Patient	PCI	Exenteration	ETPE	Aortic Clamping	Duration, Blood Loss, Transfusion	Number of Resected Organs	Number of Anastomosis	Urinary Reconstruction	Digestive Reconstruction	Empty Pelvis Management	CC/Radicality	HIPEC
1	5	SL TPE	No	no	600 min, 800 mL 2 PRBC	4	3 + rectal stump	Bricker	EC	none	CC0 / R0	Ox
2	7	SL ETPE	LC both side, pre-sacral fascia, Small bowel	Y 30 min	640 min, NA 12 PRBC, 6 FFP	5	4 + rectal stump	Bricker	EC	Breast prosthesis Left VRAM	CC0/ R0	Ox
3	8	IL ETPE	LC both side, pre-sacral fascia, Right colectomy, Small bowel	Y 19 min	660 min, 4000 mL 6 RBC, 5 FFP	6	4	Bricker	EC	none	CC0/ R0	MC
4	13	SL ETPE	LC one side, Obturator nerve, Small bowel		710 min, 2200 mL 8 PRBC, 8 FFP, 1 PC	4	4 + rectal stump	Bricker	EC	none	CC0/ R0	MC
5	16	SL ETPE	LC one side, pre-sacral fascia, Small bowel, Caecum, Ext iliac artery prosthetic replacement	no	840 min, 6000 mL 15 PRBC, 12 FFP, 2 PC	7	5 + rectal stump	Bricker	EC	none	CC0/ R1	MC
6	6	SL ETPE	LC both side, pre-sacral fascia, Small bowel	no	630 min, NA 7 PRBC, 7 FFP, 1 PC	5	4 + rectal stump	Bricker	EC	none	CC0/ R1	MC
7	13	SL ETPE	LC both side, pre-sacral fascia, Small bowel	Y 60 min	660 min, 5500 mL 7 PRBC, 4 FFP, 1 PC	5	4 + rectal stump	Bricker	EC	Biological prosthesis	CC0 /R1	MC
8	12	SL ETPE	LC one side, Ext iliac vein prosthetic replacement, Small bowel	no	600 min, 4000 mL 8 PRBC, 8 FFP, 1PC	6	0	Bilateral ureterostomy	DCAA	DCAA	CCO R0	MC
9	4	SL ETPE Splenectomy	LC one side, Face lat G, Small bowel	no	420 min, 500 mL 3 PRBC, 2 FFP	5	4	Bricker	DCAA	DCAA	CC0 R0	MC
10	4	IL TPE	No	Y 21 min	440 min, 1500 mL 4 PRBC, 2 FFP	4	0	Bilateral ureterostomy	EC	Epiplooplastie Caecum	CC0 R0	MC
11	6	SL TPE PD for PM	No	no	500 mL No transfusion	7	2 + wirsungostomy	Bilateral ureterostomy	EC	Left VRAM Caecum	CC0 R0	MC
12	12	SL TPE Small bowel caecum	No	no	580 min, 750 mL No transfusion	6	1	Bilateral ureterostomy	DCAA	DCAA Left VRAM	CC0 R0	MC
13	3	IL ETPE	Distal sacrectomy	no	420 min, 750 mL 4 PRBC, 1 FFP	4	3	Bricker	EC	Right VRAM	CC0 R0	MC
14	4	SL TPE	No	no	600 min, 800 mL No transfusion	4	3	Bricker	DCAA	DCAA Caecum	CC0 R0	MC
15	13	SL ETPE	LC both side, Small bowel, caecum Thermoablation LM	no	600 min, 2300 mL 3 PRBC, 2 FFP	6	3	Bricker	DCAA	DCAA	CC0 R0	MC
16	8	SL TPE		no	540 min, 700 mL 1 PRBC	4	0	Bilateral ureterostomy	DCAA	DCAA	CC0 R0	MC

PCI: Peritoneal Cancer Index, TPE: total pelvic exenteration, ETPE: extended total pelvic exenteration, PD: pancreaticoduodenectomy: PM: pancreas metastasis, SL: supra-levator, IL: infra-levator, LC: lateral compartment, PRBC: packed red blood cells, FFP: fresh frozen plasma, PC: platelet concentrates, DCAA: Delayed Coloanal Anastomosis, EC: end colostomy CC: completeness of cytoreduction, HIPEC: hyperthermic intraperitoneal chemotherapy, Ox: oxaliplatin, MC: mitomycin C, R0: clear resection margins, and LM: liver metastasis.

**Table 3 cancers-12-03478-t003:** Postoperative course, morbidity, recurrence, and follow-up.

Patient	ICU Stay (Day)	Hospital Stay (Day)	Complication	Number of re Laparotomy *	Dindo	Adjuvant Treatment	Recurrence	Status
1	3	48	Pelvic abscess	0	II	5FU	PM at 27 months	Dead at 41 months Sepsis following chemotherapy
2	3	39	Bricker leakage (pod 17)	1	IIIB	Pelvic Cap 50	PC at 9 months	Dead at 11 months
3	3	41	Pelvic abscess Radiologic drainage	0	IIIA	none	PC at 10 months	Dead at 13 months
4	4	53	Pelvic abscess Radiologic drainage Septic shock (urine) ICU 4 days	0	IVa	none	Vaginal recurrence at 11 months	Dead at 16 months
5	4	75	Bricker leakage (pod 23)	1	IIIB	none	PM at 29 months	Dead at 42 months
6	31	31	Bricker leakage, urosepsis and peritonitis	5	V	NR	NR	Postoperative death (31pod)
7	58 (4stay)	129	Iterative pelvis bleeding Bricker leakage, Urosepsis Fungal Peritonitis	8	V	NR	NR	Postoperative death (pod 129)
8	3	45	Urosepsis	0	II	none	LM at 12 months	Alive with LM at 44 months
9	3	30	Fluid collection left flank Radiologic drainage	0	IIIa	none		Alive at 32 months
10	3	26	Urosepsis	0	I	folfox	LM and PC at 10 months	Dead at 15 months
11	3	50	Biliary leak, Septic shock (implantable port) ICU 8 days	1	IVa	none	LM and PM at 12 months	Dead at 21 months
12	3	60	urosepsis	0	II	none	none	Alive at 19 months
13	3	39	Urosepsis Fungal septicemia	0	II	none	none	Alive at 7 months
14	3	57	Urosepsis septicemia	0	II	folfox	none	Alive at 5 months
15	3	47	Urosepsis Wound dehiscence	1	IIIb	folfox	none	Alive at 4 months
16	3	37	Urosepsis COVID 19	0	II	folfox	none	Alive at 3 months

ICU: Intensive Care Unit, pod: postoperative day, Cap 50: 50 Gy irradiation and 1600 mg/m oral capecitabine daily, PM: pulmonary metastasis, LM: liver metastasis, PC: peritoneal carcinomatosis, *: the second time of delayed coloanal anastomosis was not counted as a reoperation, Recurrence: the first recurrence site was only reported in this table, and NR: nonrelevant.

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
