# Peer review of "Total Pelvic Exenteration, Cytoreductive Surgery, and Hyperthermic Intraperitoneal Chemotherapy for Rectal Cancer with Associate Peritoneal Metastases: Surgical Strategies to Optimize Safety"

_cancers, 2020, doi:10.3390/cancers12113478_

Round 1

Reviewer 1 Report

I have read with interest this manuscript, which concerns challenging operations and treatments. I believe it is worthy to be reported since it will be of interest to the readers of Cancers.

I would suggest enhancing that we are talking about colorectal cancers and not about other malignancies. In this sense please modify the title, the summary and the abstract. These changes will help the reader to understand the research topic.

The main study limitations are the small sample size, and the absence of follow-up enough to make some conclusions from the oncological standpoint. We know that having a mortality rate of 12.5% without a proven correspondent increases in overall survival is debatable.

Author Response

We thank the reviewer for taking the time to review our manuscript. The corrections made allowed to improve the paper.

point 1: we agree with the reviewer and we have modified the title: "Total pelvic exenteration, cytoreductive surgery, and Hyperthermic Intraperitoneal Chemotherapy for rectal cancer with associate peritoneal metastases: surgical strategies to optimize safety "

 point 2: we agree with the reviewer. the limit of the small number has already been mentioned in the line 115.

 Concerning the short follow-up which does not allow a conclusion from an oncological point of view, we have softened the conclusion at the level of the abstract and the main document: "The morbidity and the mortality are high with this combination of treatment and further research is needed to assess the oncologic benefit and quality of life before such a radical approach can be recommended"

I hope we were able to respond correctly to your remarks.

Sincerely

Reviewer 2 Report

Congratulations. A great deal of work has been done and very well analyzed. Commendable determination by surgeons to overcome cancer by maximizing surgery. And here there is always a dilemma or justification for this gigantic operation. In fact, the article lacks the rationale for HIPEC, it's a positive impact in this situation. Especially when we don't know the real impact of HIPEC  to quite common small bowel conduit leakage, or septic urological complications. There is also a lack of analysis of the causes of late death. Whether it is the progression of an oncological disease, the onset of metastases, or the intensification. progression of septic complications?  In summary,  the authors demonstrate that it is possible to do CRS with HIPEC in patients with PC, but whether this is an optimization of safety remains in doubt. The article is interesting, especially considering the titanic work of surgeons to cure patients with advanced disease.

Author Response

We thank the reviewer for taking the time to review our manuscript. The corrections made allowed to improve the paper.

this is the first study on this difficult subject. it is unclear whether HIPEC had a positive effect or not. Point1: Concerning the risk of fistula on urinary reconstruction, we have shown in  patient reports  that we modify our practices with bilateral ureterostomies. point 2: we have specified in line 7 page 7 the causes of late deaths. point3: Concerning the short follow-up which does not allow a conclusion from an oncological point of view, we have softened the conclusion at the level of the abstract and the main document: "The morbidity and the mortality are high with this combination of treatment and further research is needed to assess the oncologic benefit and quality of life before such a radical approach can be recommended"  

I hope we were able to respond correctly to your remarks.

Sincerely

Reviewer 3 Report

The authors submit their manuscript entitled "Total pelvic exenteration, cytoreductive surgery and hyperthermic intraperitoneal chemotherapy; surgical strategies to optimize safety"  This is a retrospective review of 16 patients treated at a single institution over a 14 year period. The authors report a 56% incidence of severe complications with a 46 day median hospital length of stay, 37% reoperation rate and a 12.5% mortality.  They do not present any survival data.  They conclude that pelvic exenteration and HIPEC is a reasonable procedure in selected patients at expert centers.  The authors are correct in that the current literature is mainly limited to case reviews and this submitted case series is the largest I could find in the literature.  As a result there is a lack of data regarding the morbidity and mortality associated with this approach.  Therefore, the readership may find this manuscript valuable.  However, several issues need to be addressed

1) The authors had only 1 appendiceal case.  Given the recent PRODIGE 7 trial results at least in abstract form do the authors plan to continue with pelvic exoneration and HIPEC for colorectal cases?

2) How many patients had peritoneal disease outside the pelvis? 

3) Would the authors consider a staged approach such as neoadjuvant laparoscopic HIPEC then pelvic exoneration at a second stage?  What about delayed ureteral and gastrointestinal reconstruction if feasible.

4) Of the first 10 patients there are only 2 patients alive.  The last 4 patients are alive but with short follow up.  Although the authors do not report any survival outcomes, the initial prospects do not look good.

5) The summary needs softened a little.  I am not sure one can conclude that 56% severe complications and 12% mortality is reasonable without knowing the quality of life or survival outcomes.  I think the authors should conclude that the morbidity and mortality is high with the combination of HIPEC and pelvic exeneration and further research is needed to assess the oncologic benefit and the quality of life before such a radical approach can be recommended.

Author Response

We thank the reviewer for taking the time to review our manuscript. The corrections made allowed to improve the paper.

point1: this is the first study on this difficult subject. we will continue to study these patients probably in a multicentric way within Pelvex collaborative  and we will propose HIPEC to them. The prodige 7 studied the interest of Oxaliplatin regimen in addition to cytoreduction and showed that this treatment did not work. We only used oxaliplatin twice on the first 2 patients in the study. we very quickly stopped the oxaliplatin regimen because it is responsible for 7 times more bleeding than the MMC regimen.

the progide 7 study has not been published, 2 years after ASCO presentation  which raises questions. There is a lot to say about this study but this is not the place or the topic; the main one is that it only included highly selected patients. in fact over the same period in the participating centers: 1511 HIPEC ( datas from French nationwide database, PMSI) for colorectal carcinoma were performed while only 265 patients were randomized.

Point 2: all patients had peritoneal disease outside the pelvis. This point was clarified on page 4 line 89: "Detailed information on surgical resections is reported in Table 2. The median PCI was 8 (4-16) this calculation did not take into account the pelvic region; the median number of regions affected by peritoneal carcinoma was 3 (2 – 9) and the pelvic region was also excluded for this calculation."

Point 3: we do not plan to performed neoadjuvant laparoscopic HIPEC. for urinary reconstruction we plan to perform single stoma cutaneous ureterostomy. Delayed reconstruction was not plan.

point 4: we gave survival for each included patients, however we have to wait a longer follow-up period  and a multicenter study to give more solid conclusions.

Apart from the 8th patient we had no death  and the disease was severe in all cases, patient 11 had a TPE a pancreaticoduodenectomy and an HIPEC.

point 5:we agree with the reviewer.  we have softened the conclusion at the level of the abstract and the main document: "The morbidity and the mortality are high with this combination of treatment and further research is needed to assess the oncologic benefit and quality of life before such a radical approach can be recommended"

I hope we were able to respond correctly to your remarks.

Sincerely